# Harnessing Hyperbolic Geometry for Harmful Prompt Detection and Sanitization

Igor Maljkovic[1]* Maria Rosaria Briglia[2]* Iacopo Masi[2] Antonio Emanuele Cinà[1] Fabio Roli[1,3]

[1]University of Genoa, Italy     [3]University of Cagliari, Italy     [2]Sapienza University of Rome, Italy

## Abstract

Vision–Language Models (VLMs) have become essential for tasks such as image synthesis, captioning, and retrieval by aligning textual and visual information in a shared embedding space. Yet, this flexibility also makes them vulnerable to malicious prompts designed to produce unsafe content, raising critical safety concerns. Existing defenses either rely on blacklist filters, which are easily circumvented, or on heavy classifier-based systems, both of which are costly and fragile under embedding-level attacks. We address these challenges with two complementary components: Hyperbolic Prompt Espial (`HyPE`) and Hyperbolic Prompt Sanitization (`HyPS`). `HyPE` is a lightweight anomaly detector that leverages the structured geometry of hyperbolic space to model benign prompts and detect harmful ones as outliers. `HyPS` builds on this detection by applying explainable attribution methods to identify and selectively modify harmful words, neutralizing unsafe intent while preserving the original semantics of user prompts. Through extensive experiments across multiple datasets and adversarial scenarios, we prove that our framework consistently outperforms prior defenses in both detection accuracy and robustness. Together, `HyPE` and `HyPS` offer an efficient, interpretable, and resilient approach to safeguarding VLMs against malicious prompt misuse.

**Code available at**: github.com/HyPE-VLM/Hyperbolic-Prompt-Detection-and-Sanitization

**Disclaimer**: This paper contains potentially offensive text and images, included to illustrate the risks associated with VLMs and to raise awareness about their potential harmful consequences or misuse.

## 1 Introduction

Trained on massive web-scale datasets, Vision–Language Models (VLMs) have emerged as a cornerstone of modern AI. These models can jointly process and reason over visual and textual modalities, enabling a rich understanding of the semantic relationships between images and text (Nickel & Kiela, 2018). Their effectiveness stems from the ability to align linguistic and visual information within a shared embedding space, yielding robust cross-modal representations. While the idea of bridging language and vision has long been present in the research community (Joulin et al., 2016), earlier approaches were limited by the capacity of text encoders. The advent of transformer-based architectures (Vaswani et al., 2017) provided the necessary representational power, enabling VLMs to fully exploit large-scale multimodal pretraining (Radford et al., 2021). Once pretrained, these models can work as foundational components for a wide range of downstream applications, including retrieval (Radford et al., 2021; Li et al., 2022) and text-to-image generation tasks (Rombach et al., 2022; Podell et al., 2023), where pretrained text encoders are used for guiding the mapping from language to images. However, the same capabilities that make VLMs widely useful also expose them to misuse. Malicious prompts can be crafted to elicit harmful content, ranging from nudity and violence to hate speech, posing significant risks for responsible deployment (Yang et al., 2024a;c; Rando et al., 2022; Schramowski et al., 2023). Existing safeguards are limited: *blacklist-based* filters (Liu et al., 2024; Midjourney, 2025) are easily circumvented through paraphrasing or adversarial prompt optimization, while large-scale *classifier-based* systems (Li, 2025; Hanu & Unitary team, 2020) bring high computational costs and remain vulnerable to embedding-level attacks. As recent

---

*Equal contribution; order decided by dice roll. Correspondance to: `antonio.cina@unige.it`

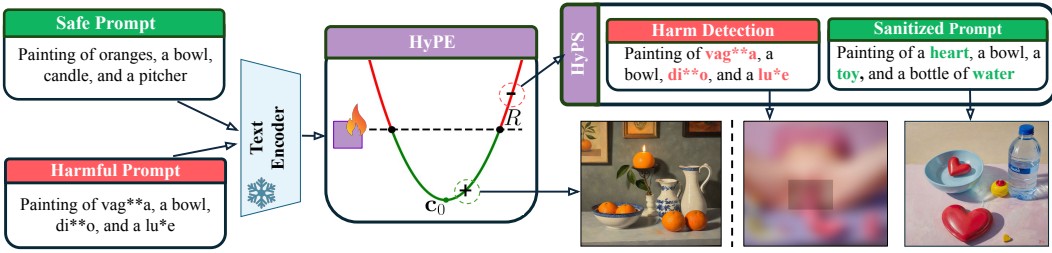

Figure 1: **HyPE and HyPS pipeline overview for T2I generation task.** User prompts are being processed by the hyperbolic frozen text encoder. Prompt classified as benign from `HyPE` are directly generated, the ones classified as malicious are then sanitized by `HyPS` before the decoding.

work on adversarial prompt manipulation has shown, even state-of-the-art filtering mechanisms fail to reliably block unsafe generations (Yang et al., 2024a;c). These limitations highlight the urgent need for lightweight, robust defenses that can detect and neutralize malicious intent beforehand.

In this work, we present a new approach for detecting and sanitizing malicious prompts in VLM pipelines. Our method builds on the structured representation properties of hyperbolic geometry (Nickel & Kiela, 2018), which naturally capture hierarchical and compositional relations in text embeddings. Specifically, we harness hyperbolic structured embeddings as a foundation and introduce two components: Hyperbolic Prompt Espial (`HyPE`), for harmful prompt detection, and Hyperbolic Prompt Sanitization (`HyPS`), for sanitization of malicious prompts. `HyPE` learns a compact region that captures the notion of *safe behavior*, effectively modeling the distribution of harmless prompts. Prompts that fall outside this learned safe region are considered anomalous and potentially harmful. Such prompts are then passed to `HyPS`, the sanitization module, which uses an explainable attribution method to identify the specific words responsible for the harmful classification. `HyPS` can then selectively modify or replace these words, neutralizing unsafe intent while preserving as much of the original semantic content as possible. An example is illustrated in Fig. 1.

We benchmark `HyPE` against five state-of-the-art detection methods across six diverse datasets. We further evaluate the robustness of our approach under a range of adversarial conditions, including MMA-Diffusion (Yang et al., 2024a), SneakyPrompt-RL (Yang et al., 2024c), StyleAttack (Qi et al., 2021), as well as a white-box adaptive attack that we introduce in this paper to explicitly target our defense. These attacks attempt to rephrase harmful inputs and manipulate their embeddings to evade harmful prompt detection systems. While existing defenses frequently fail under these manipulations, `HyPE` consistently sustains high detection performance, highlighting its robustness where prior approaches collapse. We lastly assess `HyPS` in sanitizing malicious prompts across two downstream tasks, text-to-image generation and image retrieval, showing that it can reliably neutralize harmful intent while preserving prompt semantics and enhancing the safety of VLMs.

Our contributions are threefold:

⋄ We introduce `HyPE`, a hyperbolic SVDD-based anomaly detector that identifies harmful prompts as outliers from benign distributions, while requiring training of only a single parameter.

⋄ We propose `HyPS`, an explainable sanitization mechanism that pinpoints and modifies harmful words to neutralize unsafe intent, all while preserving the original semantics of the prompt.

⋄ We conduct a comprehensive evaluation, including standard and adaptive adversarial prompt attacks, and show that `HyPE` remains effective in keeping VLMs safe.

## 2 RELATED WORK

### 2.1 VISION-LANGUAGE MODELS (VLMS)

VLMs have rapidly advanced the field of artificial intelligence by enabling systems to interpret and align visual and textual modalities jointly (Radford et al., 2021). The core mechanism of VLMs involves learning a shared embedding space in which both images and text are projected via contrastive

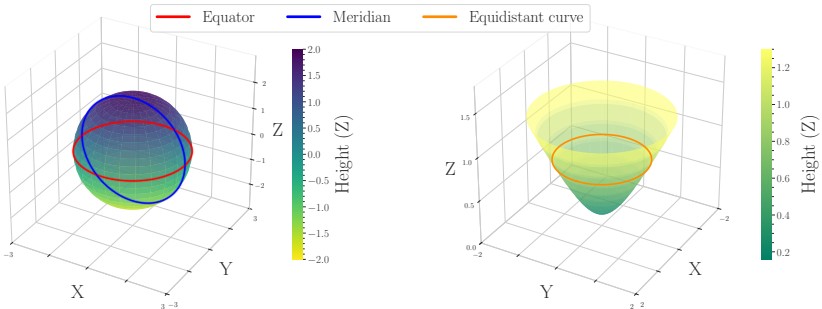

Figure 2: (*left*) SVDD hypersphere within Euclidean space. (*right*) Lorentz model upper hyperboloid. The *equidistance curve* indicates the boundary of the HSVDD hyperbolic sector.

or generative objectives, facilitating robust cross-modal understanding (Jia et al., 2021). Pioneering works such as CLIP (Radford et al., 2021), ALIGN (Jia et al., 2021), and BLIP (Li et al., 2022) leverage large-scale pretraining on noisy image-text pairs to learn rich, multimodal representations. These models demonstrate impressive capabilities across various tasks, such as image retrieval (Radford et al., 2021), visual question answering (Antol et al., 2015), image generation (Ramesh et al., 2021), and multimodal reasoning (Tan & Bansal, 2019), justifying their wide adoption.

**Hyperbolic Models.** Recent works demonstrated that hyperbolic space (Ganea et al., 2018; Peng et al., 2021) is emerging as a preferred framework for organizing structured embedding representations. Its inherent geometric properties allow modeling hierarchical and tree-like structures (Cannon et al., 1997; Krioukov et al., 2010) with minimal distortion, more effectively capturing and preserving hierarchical relationships. Hyperbolic learning has indeed been successfully applied in various domains, including few-shot learning (Gao et al., 2021; Yang et al., 2025), and VLMs (Pal et al., 2025; Peng et al., 2025). The Lorentz hyperbolic model (Nickel & Kiela, 2018; Ramasinghe et al., 2024) is commonly adopted when a hierarchical structure needs to be imposed in network learning. In an $n$-dimensional setting, the Lorentz model is defined as the upper sheet of a two-sheeted hyperboloid embedded in $(n+1)$-dimensional Minkowski space (Kosyakov, 2007). Formally, hyperbolic space $\mathbb{H}^n$ is given by:

$$\mathbb{H}^n = \{x \in \mathbb{R}^{n+1} : \langle x, x \rangle_{\mathcal{L}} = -\frac{1}{K}, \ x_0 > 0, K > 0\}, \ \text{and} \ \langle x, y \rangle_{\mathcal{L}} = -x_0 y_0 + \sum_{i=1}^{n} x_i y_i \quad (1)$$

where $\langle \cdot, \cdot \rangle_{\mathcal{L}}$ denotes the Lorentzian inner product and $K \in \mathbb{R}^+$ being a fixed positive curvature parameter. A visual representation of the Lorentz hyperboloid is shown in Fig. 2 (*right*). The hyperbolic representation provides a more structured separation space, which naturally disentangles hierarchical and compositional relations, making it well-suited for modeling data with latent hierarchical structure. Peng et al. (2025) proposes fine-tuning CLIP in hyperbolic space, achieving hierarchical alignment for open-vocabulary segmentation tasks. Hu et al. (2024) exploits hyperbolic constraints between prototypes and instances to enhance domain alignment and feature discrimination. Furthermore, Poppi et al. (2025) introduces Hyperbolic Safety Aware VLM (HySAC), which uses hyperbolic entailment loss to model the hierarchical and asymmetrical relationships between safe and unsafe image-text pairs. Lastly, Zhao et al. (2025) constructs a category-attribute-image hierarchical structure among text classes, images, and attribute prompts.

## 2.2 HARMFUL PROMPTS DETECTION

The proliferation of VLMs has also amplified their potential misuse for generating harmful, explicit, or illegal content, underscoring the need for robust safeguards to detect and filter unsafe queries. Commercial platforms such as Midjourney (Midjourney, 2025) and Leonardo.Ai already implement content filtering mechanisms as a primary line of defense. Most existing approaches formulate this task as a binary classification problem, which requires large volumes of carefully curated and annotated training data. Current methodologies typically fall into two categories: prompt-based classifiers (Li, 2025; Khader et al., 2025; Hanu & Unitary team, 2020) and embedding-based techniques (Liu et al., 2024). Despite their differences, existing implementations typically lack adapt-

ability when confronted with novel or deliberately obfuscated NSFW content, and their decision-making mechanisms often remain opaque, providing limited interpretability. In particular, conventional embedding-based approaches (Liu et al., 2024) generally treat embedding spaces as simple computational substrates, without exploiting their inherent geometric structure. In contrast, our approach reconceptualizes harmful prompt detection as an anomaly detection problem, where the geometric structure of hyperbolic space is explicitly exploited to construct a detection mechanism that is more effective, robust, and interpretable.

## 3 METHODOLOGY

In this section, we present HyPE, our detection framework for identifying and flagging harmful textual prompts. Trained exclusively on benign prompts, HyPE is based on the Hyperbolic Support Vector Data Description (HSVDD) model, which we introduce in this work by extending the traditional Support Vector Data Description (SVDD) (Tax & Duin, 2004b) to hyperbolic geometry. Once harmful prompts are detected, the system can sanitize them using a second module, namely HyPS, which highlights the words that contribute most to a prompt being classified as harmful and applies sanitization by either removing or substituting these words. Together HyPE and HyPS, illustrated in Fig. 1, are intended to safeguard VLMs, diminishing the risk of exposing to harmful content.

**Notation.** To introduce the proposed methods, we first define some common notation used throughout the following sections. We assume the existence of a tokenization algorithm $\Psi(\cdot) \in \mathbb{N}^d$, with $d = 77$, which splits an input prompt $\mathbf{p} \in \mathcal{P}$ into multiple subtokens, i.e., $\Psi(\mathbf{p}) = \{p_0, p_1, \ldots, p_d\}$. We define the hyperbolic space $\mathbb{H}^n \subset \mathbb{R}^{n+1}$ as in Eq. (1), represented using the Lorentz model, which serves as the embedding space for our approach. Lastly, we define the text encoder operating in such hyperbolic space, as in (Poppi et al., 2025), denoted by $\mathcal{T}_\theta^{\mathbb{H}}$, which, given a tokenized prompt, produces a hyperbolic embedding $\mathbf{e}_{\mathbf{p}}^{\mathbb{H}} = \mathcal{T}_\theta^{\mathbb{H}}(\Psi(\mathbf{p}))$.

### 3.1 HyPE: PROMPT DETECTION VIA ONE-CLASS HYPERBOLIC SVDD

The proposed detection defense, namely Hyperbolic Prompt Espial (HyPE), employs a hyperbolic text encoder (Poppi et al., 2025) that projects prompts into the Lorentz space. In this way, HyPE inherits a structured representation where benign prompts will form compact clusters in the resulting hyperbolic space, while harmful prompts are pushed farther away as they semantically deviate from the safe ones. We provide in Appendix A.3 and Table 7 empirical validation of this separability effect. Lastly, we design HyPE as a one-class classification head trained exclusively on benign prompts. The underlying premise is that harmful intent manifests as an outlier relative to benign behavior, making HyPE capable to flag unseen anomalous input as potentially harmful. Specifically, we extend the Support Vector Data Description (SVDD) (Tax & Duin, 2004a) unsupervised anomaly detection framework to work on the hyperbolic space. In particular, the SVDD approach works under the Euclidean geometry and it is based on the idea of learning a hypersphere that encloses the training data by jointly optimizing its center $c^* \in \mathbb{R}^d$ and radius $R^* \in \mathbb{R}$. SVDD formulation does not directly extend to hyperbolic representations, where distances are defined along geodesics rather than through simple Euclidean norms. To overcome this limitation, we extend the SVDD principle to hyperbolic space, yielding *Hyperbolic SVDD* (HSVDD). The objective for HSVDD then becomes:

$$R^* \in \underset{R}{\mathrm{argmin}} \; \frac{1}{2}R^2 + \frac{1}{n\nu}\sum_{i=1}^{n} \max\big(0, d_{\mathbb{H}}(\mathbf{p}_i, \mathbf{c_0}) - R\big), \qquad \text{with } \mathbf{c_0} = \Big[\frac{1}{\sqrt{K}}, 0, \ldots, 0\Big] \qquad (2)$$

where $\mathbf{X} = \{\mathbf{p}_1, \mathbf{p}_2, \ldots, \mathbf{p}_n\}$ is the set of training prompts, $d_{\mathbb{H}}$ denotes the pairwise geodesic distance in the Lorentz model, defined as $d_{\mathbb{H}}(\mathbf{x}, \mathbf{y}) = \frac{1}{\sqrt{K}} \mathrm{arccosh}\big(-K\langle \mathbf{x}, \mathbf{z}\rangle_{\mathcal{L}}\big)$ with $\mathbf{x}, \mathbf{z} \in \mathbb{H}^n$. Lastly, the parameter $\nu \in (0, 1]$ controls the balance between learnt volume and margin violations. When $\nu = 0$, HSVDD reduces to a pure radius minimizer, focusing solely on shrinking the hypersphere without penalizing training points that fall outside the boundary. Conversely, when $\nu = 1$, HSVDD enforces a stricter criterion by encouraging the smallest possible radius that still encloses all training samples with no violations. Unlike SVDD (Tax & Duin, 2004a), where both the center and the radius are optimized, HSVDD in HyPE fixes the center $c$ at the origin of the Lorentz model and learns only the radius $R^* \in \mathbb{R}$ as the sole parameter for detection. The optimization encourages $R$ to be as small as possible while still covering the majority of benign prompts. In this formulation, the SVDD $n$-dimensional hypersphere $\mathcal{S}^n(c, R^\star)$ with center $\mathbf{c}$ and radius $R^*$ is mapped to the

region of the hyperboloid $\mathcal{S}_{\mathbb{H}}^{n+1} \in \mathbb{R}^{n+1}$ defined as the set of points lying at a constant geodesic distance $R^*$ from the center $\mathbf{c}_0$. We illustrated this mapping in Fig. 2.

The learned hyperboloid defines the boundary of normal behavior: prompts that lie inside or close to the hyperboloid, having geodesic distance lower than $R^*$ are considered benign, while points that fall outside this boundary are treated as anomalies. Consequently, once trained, the final detection in $\texttt{HyPE}$ reduces to a simple decision rule by comparing the geodesic distance between their hyperbolic embeddings to the center $\mathbf{c}_0$ and the learned radius $R^*$. Formally, given a prompt $\mathbf{p}$ with its corresponding embedding representation in the hyperbolic geometry $\mathbf{e}_{\mathbf{p}}^{\mathbb{H}}$, $\texttt{HyPE}$ operates as follows:

$$\mathbf{HyPE}(\mathbf{p}) = \begin{cases} 0, & \text{if } d_{\mathbb{H}}(\mathbf{e}_{\mathbf{p}}^{\mathbb{H}}, \mathbf{c}_0) \leq R \\ 1, & \text{if } d_{\mathbb{H}}(\mathbf{e}_{\mathbf{p}}^{\mathbb{H}}, \mathbf{c}_0) > R \end{cases} \tag{3}$$

where class 0 corresponds to a $\texttt{Safe}$ prompt and class 1 corresponds to a $\texttt{Harmful}$ prompt.

## 3.2 HYPS: HYPERBOLIC PROMPT SANITIZATION

Prompt sanitization (Chong et al., 2024) aims to identify and modify harmful words in user-provided prompts before downstream VLMs process them. The goal is to prevent malicious content generation while at the same time preserving the utility of the prompt. In our framework, sanitization is implemented in a second module, namely $\texttt{HyPS}$, which builds directly on the predictions from $\texttt{HyPE}$. Specifically, once harmful prompts are detected by $\texttt{HyPE}$, $\texttt{HyPS}$ is then used to explain the model's decision using a post-hoc explanation technique (Madsen et al., 2022) that highlights the tokens most responsible for a harmful classification. This attribution step serves two purposes. On the one hand, it identifies the specific words that drive the detector's prediction, guiding the sanitization process. On the other hand, it provides a sanity check to ensure that the model is not relying on spurious correlations when flagging prompts as unsafe. Formally, given the tokenizer $\Psi$ and $\texttt{HyPE}$ detector, the post-hoc explanation algorithm $\phi$ computes an attribution vector for a prompt $\mathbf{p}$:

$$\phi\big(\Psi(\mathbf{p}), \texttt{HyPE}\big) = (a_1, a_2, \ldots, a_d), \tag{4}$$

where $a_i \in \mathbb{R}$ measures the influence of token $p_i$ on the detector's decision. In our work, we quantify token-level contributions using Layer Integrated Gradients (Sundararajan et al., 2017). Furthermore, because modern transformer-based models process text as subword tokens rather than whole words (Vaswani et al., 2017), we aggregate token-level attribution scores into word-level ones. If a word is split into multiple tokens, the attribution scores of its constituent tokens are summed to obtain a single influence score. This ensures that each word in the original prompt receives a coherent importance score, which can be directly interpreted by humans and used to guide sanitization. Once harmful words are identified, $\texttt{HyPS}$ applies a sanitization algorithm to neutralize unsafe intent while preserving as much of the original meaning as possible. We experiment with three sanitization strategies of increasing sophistication designed to neutralize the words that contribute most to the harmful prediction by $\texttt{HyPE}$, effectively removing elements that could drive harmful content generation or retrieval. In the following paragraphs, we describe each sanitization strategy in detail.

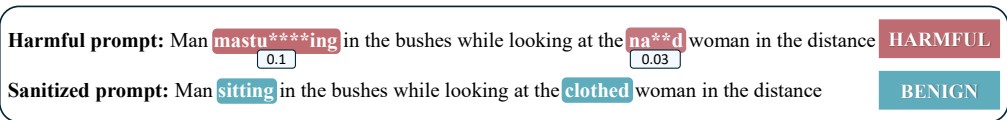

Figure 3: Harmful Prompt Sanitization via $\texttt{Thesaurus+LLM}$ ($\texttt{HyPS}$).

**Word Removal.** It removes the most influential words identified by $\phi$, i.e., those that contribute most strongly to a harmful prediction by $\texttt{HyPE}$. This strategy ensures maximum reduction of harmful content but comes at the expense of prompt coherence and informativeness.

**Thesaurus + Word Removal.** With this approach, harmful words are first replaced with antonyms obtained from the open-source Merriam Thesaurus API.[1] If no suitable antonym is found, the word is removed. This method reduces semantic loss compared to direct removal while better preserving the intent of the original prompt. When multiple antonyms are present, the one with the highest CLIP similarity (Radford et al., 2021) compared to the original harmful word is chosen.

---

[1] https://www.merriam-webster.com/

**Thesaurus + LLM.** To further improve semantic preservation, we extend the previous strategy by incorporating an instruction-tuned LLM, Qwen3-14B (Team, 2025)—see instruction details in Appendix A.5. When no suitable antonym is available, the LLM generates a safe replacement instead of simply discarding the word. As a result, this technique maximizes semantic preservation compared to prior methods. In Fig. 3, we demonstrate the effectiveness of the `Thesaurus+LLM` approach in sanitizing a harmful prompt while preserving its original semantics. In this example, the word "*naked*" would be substituted with its corresponding antonym "*clothed*", while the word "*masturbating*", having no antonyms, has been changed to the safe word "*sitting*" using the LLM.

## 4 EXPERIMENTS

We report an extensive experimental evaluation of `HyPE` and `HyPS` across six datasets, two adversarial attack settings, and two downstream tasks, comparing against state-of-the-art detection methods and demonstrating consistent improvements in both detection rate and semantic preservation.

### 4.1 EXPERIMENTAL SETUP

**Datasets.** We evaluate `HyPE` on 6 different datasets of naturally occurring prompts, grouped into two main categories: *Paired Prompts* and *Single-Class Prompts*. The former category of datasets, including ViSU (Poppi et al., 2024), MMA (Yang et al., 2024a), and SneakyPrompt (Yang et al., 2024c), provide paired examples of safe and harmful prompts with closely matched semantics, allowing us to evaluate the model's ability to detect subtle differences in intent. Complementary, *Single-Class Prompts* datasets consist exclusively of either safe or harmful prompts and are used to test models under imbalanced settings. Within this category, we find COCO (Lin et al., 2014) (safe only), I2P* (Schramowski et al., 2023) (harmful only), and NSFW56K (Li et al., 2024) (harmful only). Further details about the dataset composition are provided in Appendix A.2.

**Adversarial Prompts.** We further assess the performance of `HyPE` against state-of-the-art detectors on adversarial prompts deliberately crafted to evade safety filters by obfuscating or disguising harmful intent (Chu et al., 2024). To this end, we consider two recent adversarial attacks: MMA (Yang et al., 2024a) and SneakyPrompt-RL (Yang et al., 2024c). For MMA, the corresponding adversarial prompts are generated by iteratively modifying a random suffix until its embedding aligns with that of a harmful target prompt, thereby concealing its malicious intent. We run the MMA attack in a white-box setting directly against the HySAC text encoder (Poppi et al., 2025), and we refer to the resulting dataset with adv-MMA. For SneakyPrompt-RL (Yang et al., 2024c), we adopt the default attack setup targeting Stable Diffusion (Rombach et al., 2022) and apply it to the ViSU dataset, where sensitive words are replaced with short subword tokens until the prompt is no longer flagged as unsafe by the text_match safety filter. We refer to the resulting dataset as adv-ViSU.

**Adaptive Attacks.** To evaluate `HyPE` under adaptive scenarios, we consider two attacks. The first is an adaptive version of StyleAttack (Qi et al., 2021), which paraphrases prompts to evade detection by querying each model individually, generating model-specific adversarial paraphrases. Further details and results for this attack are available in Appendix A.9. The second is a custom adaptive attack we propose to extend the MMA-Diffusion (Yang et al., 2024a) to explicitly target `HyPE` under a worst-case adaptive scenario. Under this attack we consider a strong attacker that has full access to the defense, including the hyperbolic encoder $\mathcal{T}_\theta^{\mathbb{H}}$ and the learned HSVDD decision boundary defined by the center $\mathbf{c}$ and radius $R$. Given a harmful target prompt $\mathbf{p}_T \in \mathcal{P}$, the attacker optimizes a candidate prompt $\mathbf{p}_C \in \mathcal{P}$ maximizing the semantic similarity with $\mathbf{p}_T$ while remaining within the benign hyperbolic region. Formally, we define the adversarial optimization problem as:

$$\mathbf{p}_C^\star = \arg \max_{\mathbf{p}_C \in \mathcal{P}} S_{\cos}\left(\mathbf{e}_{\mathbf{p}_C}^{\mathbb{H}}, \mathbf{e}_{\mathbf{p}_T}^{\mathbb{H}}\right) - \lambda \max\left\{0,\ d_{\mathcal{L}}(\mathbf{c}, \mathbf{e}_{\mathbf{p}_C}^{\mathbb{H}}) - R\right\}, \tag{5}$$

where $\mathbf{e}_{\mathbf{p}_C}^{\mathbb{H}} = \mathcal{T}_\theta^{\mathbb{H}}(\Psi(\mathbf{p}_C))$, $\mathbf{e}_{\mathbf{p}_T}^{\mathbb{H}} = \mathcal{T}_\theta^{\mathbb{H}}(\Psi(\mathbf{p}_T))$ are the hyperbolic embeddings of the candidate and target prompts, , and $S_{\cos}(\cdot, \cdot)$ denotes the cosine similarity between them. The ReLU-style term $\max\{0, d_{\mathcal{L}}(\mathbf{c}, \mathbf{e}_{\mathbf{p}_C}^{\mathbb{H}}) - R\}$ penalizes embeddings outside the learned decision boundary. The parameter $\lambda \in [0, 1]$ controls the importance of the attacker to evade detection relative to preserving semantic similarity with the target prompt. Specifically, as $\lambda$ increases, the attack prioritizes evading `HyPE` by generating candidate prompts that lie within the learned benign region of radius $R$. Additional details on this attack and its implementation are provided in Appendix A.10.

**HyPE and HyPS Configuration.** `HyPE` implements the anomaly detection adopting the HSVDD framework. In particular, to implement the hyperbolic deep input preprocessing layer, we leverage the pretrained text encoder of HySAC (Poppi et al., 2025) model. `HyPE` detection module is trained only on the benign prompts from ViSU, following Eq. (2) and optimized by setting $\nu = 0.0325$. An ablation study on this hyperparameter is provided in Appendix A.7. The explanation method used by `HyPS` for inspecting harmful prompts detected by `HyPE` is Layer Integrated Gradients (LIG) (Sundararajan et al., 2017). We adapt LIG to operate on the embedding layer of the HySAC transformer-based text encoder, attributing `HyPE`'s output directly to the token embeddings obtained from the first layer, an interpretable stage where each embedding corresponds one-to-one with an input token. We then compute the gradients of the `HyPE`'s output with respect to the token embeddings, and by accumulating them, we obtain attribution scores that capture each token's influence. Finally, token-level scores are aggregated into word-level attributions to guide the sanitization step.

**Downstream Tasks.** Being developed to support VLMs, `HyPE` and `HyPS` serve as plug-and-play protection mechanisms that can be applied across different application scenarios. In this work, we evaluate their effectiveness in two practical tasks: Text-to-Image (T2I) generation and Image Retrieval (IR). For the T2I task, we use the Stable Diffusion (SD) pipeline. Our goal is to detect harmful intent in prompts from the ViSU dataset, sanitize them, and then compare the generated outputs to verify that unsafe content is removed while semantic intent is preserved. To ensure that malicious prompts would otherwise lead to unsafe results, we adopt a standard SD pipeline with a decoder configuration known to produce realistic NSFW content when given harmful inputs.[2] For the IR task, we leverage the joint embedding space of VLMs, which enables cross-modal retrieval by aligning text and image representations. Given a prompt $\mathbf{p}$ and a pool of $m$ candidate images $I = \{i_j\}_{j=1}^m$, IR is performed by computing the cosine similarity between the embedding of $\mathbf{p}$ and the embedding of each candidate image $i_j \in I$. Lastly, candidate images are then ranked according to their similarity to the input prompt $\mathbf{p}$, and the top-$k$ results are returned as those most semantically aligned with the input query. For evaluation in the IR setting, we use the UnsafeBench dataset (Qu et al., 2024) containing paired malicious and benign prompts with their corresponding images, and measure how `HyPE` and `HyPS` improve retrieval safety by detecting and sanitizing harmful queries before retrieval. Across both downstream task goal is to showcase how `HyPE` and `HyPS` enabling the VLMs to prevent harm return semantically relevant yet safe outputs.

**Evaluation Metrics.** For the detection task, we report precision, recall, and the F1 score (Powers, 2020). Precision measures the proportion of prompts identified as harmful that are indeed harmful, while recall captures the proportion of truly harmful prompts that are correctly detected. The F1 score, defined as the harmonic mean of precision and recall, provides a single measure that balances these two aspects. In our context, high precision indicates a low number of false positives, whereas high recall reflects the effective detection of harmful prompts. For single-class datasets, where only safe or harmful prompts are present, we report the classification Accuracy (`Acc.`). For downstream applications, we adopt task-specific metrics. In T2I generation, we use CLIPScore (Hessel et al., 2021) to evaluate the semantic alignment between generated images and their conditioning prompts. In image retrieval, we report Recall@k (`R@k`) (Manning, 2008), which measures the fraction of relevant images retrieved among the top-$k$ results, and Safe@k (`S@k`), which quantifies the proportion of retrieved images that are safe. Finally, to evaluate the semantic consistency between the original harmful prompts and their sanitized counterparts, we compute the cosine similarity using both SBERT (Reimers & Gurevych, 2019) and CLIP embeddings (Radford et al., 2021).

## 4.2 EXPERIMENTAL RESULTS

**Harmful Prompt Detection.** We compare `HyPE` against state-of-the-art classifiers, including NSFW Classifier (Li, 2025), DiffGuard (Khader et al., 2025), Detoxify (Hanu & Unitary team, 2020), LatentGuard (Liu et al., 2024), and GuardT2I (Yang et al., 2024b). Table 1 reports precision, recall , and F1 scores for paired, single-class, and adversarial prompt datasets. Notably, despite being trained only on benign training samples from the ViSU dataset, `HyPE` consistently achieves the highest F1 scores across all datasets, suggesting a strong generalization capacity and reliable performance across both harmful and benign prompts. More specifically, `HyPE` achieves the highest F1 scores on ViSU (0.98), MMA (0.95), showing balanced precision and recall in contrast to other models that exhibit extreme behavior, such as Detoxify achieving 0.98 precision but only 0.26 recall

---

[2] We use `stablediffusionapi/newrealityxl-global-nsfw` available on HuggingFace.

on ViSU, or NSFW-Classifier attaining $0.96$ recall on MMA but only $0.75$ F1 due to lower precision. On the SneakyPrompt dataset, `HyPE` achieves the second highest recall score ($0.93$) after GuardT2I, while maintaining a higher precision of $0.68$, illustrating its ability to detect subtle harmful variations without excessive false positives. In single-class datasets, `HyPE` demonstrates strong detection of harmful prompts, achieving $0.99$ accuracy on NSFW56k and $0.66$ on I2P*, while maintaining $0.99$ accuracy on benign COCO prompts. Lastly, when considering adversarially crafted prompts, we observe again how `HyPE` maintains the highest overall F1, scoring $0.96$ on adv-MMA and $0.80$ on adv-ViSU, despite other models occasionally achieving slightly higher precision or recall in isolation. These results highlight that `HyPE` is not only highly effective on naturally occurring harmful prompts but also robust against adversarially optimized ones, consistently delivering balanced detection of harmful and benign prompts across diverse datasets, including those not seen during training.

Table 1: Comparison for harmful prompt detection on paired, single-class, and adversarial datasets.

| Method | *Paired Prompts* | | | | | | | | | *Single-class Prompts* | | | *Adversarial Prompts* | | | | | |
| | **ViSU** | | | **MMA** | | | **SneakyPrompt** | | | **COCO** | **I2P*** | **NSFW56k** | **adv-MMA** | | | **adv-ViSU** | | |
| | Pr | Rec | F1 | Pr | Rec | F1 | Pr | Rec | F1 | Acc | Acc | Acc | Pr | Rec | F1 | Pr | Rec | F1 |
|---|---|---|---|---|---|---|---|---|---|---|---|---|---|---|---|---|---|---|
| NSFW-Classifier | 0.70 | 0.80 | 0.75 | 0.61 | **0.96** | 0.75 | 0.67 | 0.93 | **0.78** | 0.61 | 0.65 | 0.95 | 0.62 | **0.99** | 0.76 | 0.62 | 0.65 | 0.64 |
| DiffGuard | 0.27 | 0.36 | 0.31 | 0.47 | 0.88 | 0.61 | 0.46 | 0.85 | 0.60 | **0.99** | 0.28 | 0.89 | 0.89 | 0.97 | 0.93 | 0.97 | 0.40 | 0.65 |
| Detoxify (Orig) | **0.98** | 0.26 | 0.40 | 0.96 | 0.88 | 0.92 | **1.00** | 0.28 | 0.44 | **0.99** | 0.03 | 0.34 | 0.93 | 0.56 | 0.70 | **1.00** | 0.07 | 0.13 |
| Latent Guard | 0.79 | 0.52 | 0.63 | 0.95 | 0.81 | 0.88 | 0.91 | 0.41 | 0.57 | 0.84 | 0.35 | 0.52 | 0.94 | 0.80 | 0.86 | 0.49 | 0.18 | 0.27 |
| GuardT2I | 0.48 | 0.77 | 0.59 | 0.58 | 0.92 | 0.72 | 0.52 | **0.95** | 0.66 | 0.77 | 0.26 | 0.09 | **1.00** | 0.10 | 0.19 | 0.42 | **0.71** | 0.53 |
| **HyPE (Ours)** | **0.98** | **0.98** | **0.98** | **0.98** | 0.92 | **0.95** | 0.68 | 0.93 | **0.78** | **0.99** | **0.66** | **0.99** | 0.98 | 0.93 | **0.96** | 0.97 | 0.67 | **0.80** |

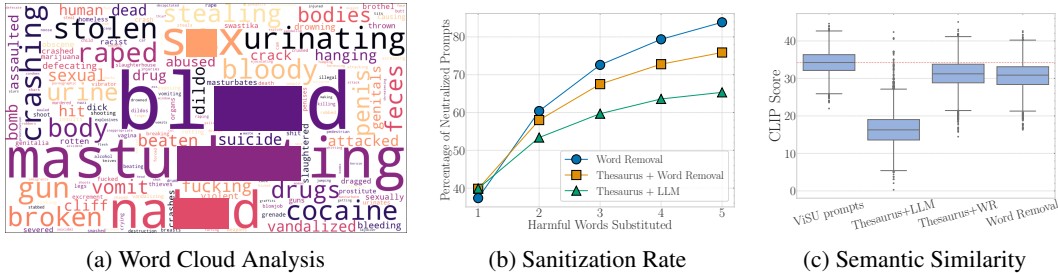

(a) Word Cloud Analysis     (b) Sanitization Rate     (c) Semantic Similarity

Figure 4: Harmful prompt sanitization analysis.

**Harmful Prompt Sanitization.** We present the sanitization performance of `HyPS` applied to harmful prompts detected by `HyPE`, with the objective of removing malicious intent while preserving semantic content. To this end, considering the ViSU test prompts detected as harmful by `HyPE`, we first illustrate in Fig. 4a a word cloud of the most relevant words identified by `HyPS`. Specifically, for each harmful prompt, we consider only the most relevant words identified by the explanation method $\phi$ in `HyPS`, and we aggregate the frequencies of these words across the dataset. This visualization demonstrates that `HyPS` consistently identifies meaningful, harmful words rather than relying on spurious correlations. For a more detailed analysis of the word cloud, see Appendix A.4. We evaluate the effectiveness of the three sanitization strategies in `HyPS` by measuring the percentage of prompts reclassified as benign by `HyPE` after sanitization. The results in Fig. 4b show that across methods, a substantial portion of harmful prompts are successfully neutralized, meaning `HyPE` no longer flags them as malicious, with rates ranging from $65\%$ (`Thesaurus+LLM`) to $85\%$ (`Word Removal`). Although `Word Removal` achieves the highest neutralization, this comes at the cost of semantic preservation. In particular, `Thesaurus+LLM` modifies only harmful words, preserving the original prompt meaning, while `Word Removal` removes these elements and therefore loses more semantic content. Quantitative evaluation using SBERT and CLIP embeddings confirms indeed that prompts sanitized with `Thesaurus+LLM` remain highly similar to the originals, with mean cosine similarities of $0.82$ and $0.87$, respectively, indicating that harmful elements can be removed without compromising the user's intent. The `Thesaurus+Word Removal` method provides an intermediate balance, neutralizing a moderate fraction of prompts while incurring noticeable semantic loss. Overall, `Word Removal` is most effective for complete prompt neutralization, `Thesaurus+LLM` is preferable when preserving prompt semantics is necessary, and `Thesaurus+Word Removal` offers a compromise solution between them.

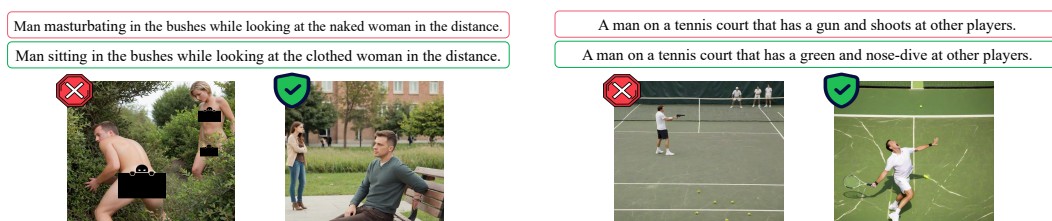

Figure 5: Qualitative comparison for the T2I task. Images generated with SD-XL using sanitized prompts (green) do not exhibit harmful content, while retaining the original prompt context (red).

**T2I Generation Task.** Following the setup described in Section 4.1, we incorporate `HyPE` and `HyPS` into the Stable Diffusion (SD) pipeline as a plug-and-play prompt moderation module, aiming to prevent the SD from generating harmful content. In this setting, we consider harmful prompts from the ViSU dataset, and performance is evaluated using both qualitative and quantitative assessments on the generated images. Fig. 5 shows two qualitative examples. Red rectangles indicate harmful prompts, whose unfiltered generations correspond to images flagged as malicious. Green rectangles show the sanitized prompts produced by `Thesaurus+LLM` and their paired images, flagged as safe. Notably, images generated with SD using these sanitized prompts are deprived of the malicious content while preserving the original prompt context. Complementary, to quantitatively measure the effectiveness of moderation, we generate images for all harmful ViSU prompts both without filtering and with `HyPE` and `HyPS`, using each sanitization method. We then compute CLIPScore between each generated image and its corresponding original malicious description to assess how much the generated content deviates from the initial harmful description. Fig. 4c shows that `Thesaurus+LLM` yields lower CLIP scores against the malicious prompt, indicating its effectiveness in reducing alignment with harmful content while preserving semantic coherence.

| Prompts | R@1 | S@1 | R@5 | S@5 |
|---|---|---|---|---|
| **Harmful prompts** | 39.49 | 0.0 | 72.23 | 0.0 |
| Word Removal | 6.91 | 49.34 | 20.96 | 44.04 |
| Thesaurus+Word Removal | 7.02 | 49.00 | 20.90 | 44.07 |
| Thesaurus+LLM | 7.08 | 49.29 | 21.07 | 44.19 |

Table 2: Detection results for the IR downstream task. S@k measures the proportion of retrieved images that are safe within the top-$k$ results.

Figure 6: Histogram of NudeNet detections.

**Image Retrieval Task.** For the IR task, we rely on the UnsafeBench dataset, which provides a large collection of images and paired captions, covering both safe and unsafe concepts. From this dataset, we select a subset of 3,702 unsafe captions and use them to retrieve top-$k$ images with the CLIP (Radford et al., 2021) encoder. We then repeat the retrieval process but using sanitized prompts generated by the three sanitization strategies in `HyPS`. As shown in Table 3, we evaluate performance for $k = 1, 5$ using two metrics: `R@k` and `S@k`. Results confirm that sanitization substantially reduces the likelihood of retrieving images aligned with harmful prompts, while increasing the chance of retrieving images aligned with safe concepts. A qualitative example is depicted in Fig. 7, where the retrieval results for a harmful prompt (*left*) are compared with those for its sanitized counterpart, which contains no harmful content, generated by `Thesaurus+LLM` (*right*).

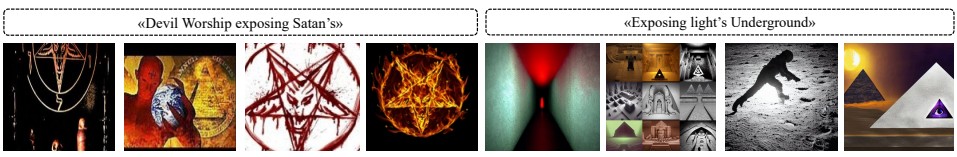

Figure 7: Qualitative evaluation of the top-4 images retrieved when prompting: *left*, the UnsafeBench harmful prompt; *right*, the corresponding sanitized prompt by `Thesaurus+LLM`.

Table 3: `HyPE` performance under the adaptive attack at increasing $\lambda$.

| $\lambda$ | Pr | Rec | F1 |
|---|---|---|---|
| 0 | 0.98 | 0.99 | 0.98 |
| 0.1 | 0.98 | 0.97 | 0.98 |
| 0.3 | 0.98 | 0.80 | 0.88 |
| 0.5 | 0.95 | 0.35 | 0.51 |
| 0.7 | 0.91 | 0.18 | 0.31 |
| 1 | 0.87 | 0.12 | 0.22 |

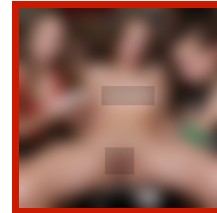 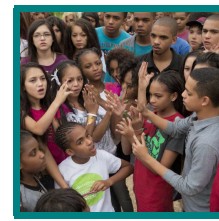

Figure 8: The left image illustrates a sample generated by an adaptive attack with $\lambda = 0.1$, while the right image shows the result of the same attack with $\lambda$ increased to 1.

**Concept Combination Attack.** We lastly evaluate the detection capabilities of `HyPE` against the Concept Combination Attack (CCA) (Petsiuk & Saenko, 2024), an embedding-level attack that we adapt from Euclidean to hyperbolic space. Full details of its implementation are provided in Appendix A.6. Our goal is to show that while the hyperbolic encoder is vulnerable to these attacks, `HyPE` can successfully protect it. We test CCA on the HySAC encoder within the SD pipeline and confirm, through embedding alignment with the CLIP text encoder in Appendix A.3, that HySAC is suitable for this generative pipeline. We then let SD generate images from ViSU safe prompts under three conditions: unattacked, filtered by `HyPE`, and attacked via CCA. Following the CCA setting, injected concepts include *"text reading nudity f**k n**ed n**e p**n bl**d"* and removed concepts include *"text, written"*. Finally, to identify any exposed body parts in the generated images, we use NudeNet (Mandic et al., 2024). Fig. 6 shows a histogram of detected occurrences for each setting. Results reveal that the HySAC is highly vulnerable to CCA, with a sharp increase in unsafe image outputs. Conversely, `HyPE` provides strong defense, effectively reducing unsafe content to zero.

**Adaptive Attack.** We evaluate `HyPE` under a custom adaptive attack following the formulation in Eq. (5) on the ViSU dataset, using 1,000 randomly-chosen prompts. The attack is controlled by a parameter $\lambda$ that balances the attacker's objective of evading detection with preserving the harmful intent of the prompt. We gradually increase $\lambda$ from 0 to 1 and observe from Table 3 two main trends: (i) for small $\lambda$, `HyPE` successfully detects adversarial prompts; (ii) as $\lambda$ approaches 1, the attack can evade detection more effectively. However, it is important to note that higher values of $\lambda$ also lead to substantial changes in the prompts, significantly removing or severely reducing their malicious intent (see examples in Table 11). Furthermore, these findings are reinforced by additional qualitative analyses shown in Fig. 8, with Appendix A.10 providing a richer set of examples. Using the T2I pipeline, we generate images from adaptive prompts to assess the effects of the attack on a generative task. The analysis shows that as $\lambda$ decreases, the harmfulness of the generated images increases, which is counterbalanced by improved model performance provided in Table 3. These results reveal a fundamental trade-off in adaptive attacks: evading `HyPE` requires sacrificing the harmfulness intent in the prompts. Overall, the results indicate that `HyPE` reliably detects adversarial prompts as long as they retain their malicious intent, evidencing its robustness even when considering worst-case adaptive scenarios.

## 5 CONCLUSION

We introduced `HyPE` and `HyPS`, two complementary modules for detecting and sanitizing harmful prompts. `HyPE`, trained on benign data only, leverages hyperbolic SVDD to identify malicious prompts as outliers, while `HyPS` uses explainable attributions to select and neutralize harmful words without sacrificing semantic consistency. Our extensive evaluation, across four datasets, four adversarial scenarios, and two downstream tasks, shows that `HyPE` consistently outperforms state-of-the-art detectors, achieving balanced precision, recall, and F1 scores. Furthermore, CCA experiments confirm that `HyPE` provides robust protection even against embedding-level attacks. We also observe a key trade-off in adaptive attacks, where attempts to evade the defense typically succeed only when the malicious intent in the prompts is removed or severely reduced. Lastly, we demonstrate that `HyPS` effectively sanitizes harmful inputs, preserving the original semantic intent while preventing unsafe outputs. Collectively, our results indicate that `HyPE` and `HyPS` together offer an effective, generalizable, and explainable solution for improving the safety of VLMs.

**Acknowledgments.** This work has been partially supported by project FISA-2023-00128 funded by the MUR program "Fondo italiano per le scienze applicate"; the European Union's Horizon Europe research and innovation program under the project ELSA, grant agreement No 101070617; by EU - NGEU National Sustainable Mobility Center (CN00000023) Italian Ministry of University and Research Decree n. 1033—17/06/2022 (Spoke 10); by projects SERICS (PE00000014) and FAIR (PE0000013) under the NRRP MUR program funded by the EU - NGEU; by PRIN 2022 project 20227YET9B "AdVVent" CUP code B53D23012830006, and project BEAT (Better dEep leArning securiTy), a Sapienza University project. Lastly, we acknowledge the EuroHPC Joint Undertaking for awarding this project access to the EuroHPC supercomputer LEONARDO, hosted by CINECA (Italy) and the LEONARDO consortium through an EuroHPC Development Access call.

**Ethics Statement.** Based on our comprehensive analysis, we assert that this work does not raise identifiable ethical concerns or foreseeable negative societal consequences within the scope of our study. On the contrary, our contributions aim to enhance the safety of vision-language models by improving the detection and sanitization of harmful prompts.

**Reproducibility.** To ensure reproducibility, we provide a detailed description of our experimental setup in Section 4.1, including datasets, models, and adversarial attacks, along with their sources. Furthermore, our source code has been included as part of the supplementary material and is available at https://github.com/HyPE-VLM/Hyperbolic-Prompt-Detection-and-Sanitization .

**LLM Usage.** Large language models were used exclusively for text polishing and minor exposition refinements. All substantive research content, methodology, and scientific conclusions were developed entirely by the authors.

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
