# OpenReview forum: "Harnessing Hyperbolic Geometry for Harmful Prompt Detection and Sanitization"
_ICLR.cc/2026/Conference — ICLR 2026 Poster_

### Official Review · Reviewer_WR4w · 2025-10-31

**Soundness:** 3
**Presentation:** 3
**Contribution:** 3
**Rating:** 6
**Confidence:** 4

**Summary:**

The paper introduces the HyPE and HyPS, a system for malicious prompt detection and sanitization based on constructing prompt embeddings in a hyperbolic space that naturally separates malicious from benign prompts. Sanitization is performed by selectively replacing harmful words in the prompt using an explainable attribution method. The approach is benchmarked against a range of SOTA approaches and stress-tested against some adversarial attacks.

**Strengths:**

* Notwithstanding my questions below, the hyperbolic approach is well-suited to single-class training, reducing a dependency on labeled malicious examples.
* The method includes a sanitization step, enabling end-to-end workflows where refusal is undesirable.
* Empirical experimental results are relatively thorough and demonstrate strong performance of the method compared to several baselines.

**Weaknesses:**

* Section 3.1 appears to leave out important details of how the mapping from token sequence to hyperbolic space is learned, instead focusing on how R is tuned. It appears that the experiments leverage a pretrained encoder for this step (HySAC).
* A different encoder is selected for HyPS and this should be justified.

**Questions:**

Explain in more detail the reasoning behind the choice of a hyperbolic space. You say it "naturally disentangles  hierarchical and compositional relations, making it well-suited for modeling data with latent hierarchical structure". What is the latent hierarchical structure in text-to-image prompts? Is the expectation that the malicious/benign label represents the top level of the hierarchy?

The underlying assumption appears to be that malicious prompts will be outliers in this learned space- have you stress-tested your approach using OOD benign images?

Minor notation issue in Eq 1: if x is n+1 dimensional then the Lorentzian inner product should only sum up to n (the elements of x are indexed 0 to n, comprising n+1 elements).

For HyPS: How do you make the decision about what words to replace (is it a hard threshold or more adaptive method)?

Suggest blurring the offensive image in Fig 1- readers will trust your claim that it is offensive.

Detection is trained using HySAC embeddings but attribution appears to be applied using CLIP embeddings and I'm curious to better understand the implications. Aside from empirical validation, can you support the claim that the two embedding spaces can be distinct? Is it difficult to implement LIG against the HySAC encoder?

Presentation: Table 1 misrepresents HyPE as the winning method in the Precision column for adv-MMA (should be GuardT2I). There are some ties that should also be marked- eg SneakyPrompt recall and COCO accuracy.  Ideally you would also indicate statistically significant wins.

For HyPS experiments, do you also validate that the rewritten prompts fall inside R?

---

> ### Author Response · Authors · 2025-11-20
> **Clarifications on hyperbolic geometry**
>
> **Mapping from token sequence to hyperbolic space**
> > Section 3.1 appears to leave out important details of how the mapping from token sequence to hyperbolic space is learned, instead focusing on how R is tuned. It appears that the experiments leverage a pretrained encoder for this step (HySAC).
>
> We thank the reviewer for the constructive feedback. We clarify that the mapping from token sequences to hyperbolic space is performed using a pretrained hyperbolic encoder, HySAC, which is explicitly trained to capture both benign and harmful prompt embeddings. To provide further insight into this representation, we have added a UMAP visualization of the token embeddings in hyperbolic space, highlighting the clear separation between benign and malicious prompts. Please refer to **Appendix A.3** (**Figure 9**) for these visualizations, which illustrate how HySAC effectively organizes the embedding space and supports the anomaly detection performed by HSVDD.
>
> ---
> ---
>
> **Typing error**
> > A different encoder is selected for HyPS, and this should be justified.
>
> We are grateful to the reviewer for noticing this misunderstanding, caused by a typing error on our side. We clarify that HyPS uses the same text encoder as our anomaly detector HyPE, namely HySAC. Since HyPS relies on the predictions and attributions produced by HyPE, using the same encoder ensures full compatibility between the two modules. Both models operate on the same hyperbolic embeddings, and this design choice is reflected in the code provided with the submission.
>
>
> ---
> ---
>
>
> **Hyperbolic SVDD over Euclidean SVDD**
> > Explain in more detail the reasoning behind the choice of a hyperbolic space. You say it "naturally disentangles hierarchical and compositional relations, making it well-suited for modeling data with latent hierarchical structure". What is the latent hierarchical structure in text-to-image prompts? Is the expectation that the malicious/benign label represents the top level of the hierarchy?
>
> > can you support the claim that the two embedding spaces can be distinct?
>
> We sincerely thank the reviewer for this insightful question. Our choice of hyperbolic geometry is motivated by the fact that harmful intent in text prompts typically arises not from a single token but from compositions of lower-level modifiers (e.g., violent, bomb, etc.). Hyperbolic space enables the construction of hierarchically organized embedding spaces that help the clustering of the harmful and safe concepts. Therefore, it is well-suited for modeling such structures, allocating exponentially more volume as one moves away from the origin, enabling natural separation between core semantics and increasingly specific (and potentially harmful) refinements of the prompt.
> To support this, we revised the paper to include new UMAP visualizations in **Appendix A.3 (Figure 9)** showing that hyperbolic embeddings preserve this hierarchical organization and clearly distinguish benign from harmful prompts. Quantitative comparisons in **Table 5** and **Table 7** demonstrate that HSVDD leverages this structure for superior detection performance, outperforming Euclidean SVDD, which struggles to capture the same hierarchy.
>
> | Method      | Pr    | Rec   | F1    |
> |-------------|-------|-------|-------|
> | CLIP-SVDD   | 0.08  | 0.96  | 0.66  |
> | Hype        | **0.98** | **0.98** | **0.98** |
>
> These results therefore reinforce both the theoretical rationale and empirical effectiveness of using hyperbolic embeddings for the considered harmful prompt detection task.
>
> We have revised the paper to clarify this discussion.

---

> ### Author Response · Authors · 2025-11-20
> **Questions**
>
> **Q1.**
> > The underlying assumption appears to be that malicious prompts will be outliers in this learned space- have you stress-tested your approach using OOD benign images?
>
> We thank the reviewer for raising this important point. Since HyPE is trained exclusively on benign ViSU prompts, any evaluation on benign prompts originating from different data sources already constitutes an OOD setting. To stress-test this, we evaluate HyPE on 30,000 COCO captions (see **Table 1**), which differ substantially in structure and style from the ViSU data. These captions are therefore OOD benign samples, and we can notably observe from the resulting table that HyPE maintains strong performance (i.e., 99% of accuracy), indicating that the learned decision boundary does not overfit to the training distribution.
> In addition, we evaluate HyPE on two more paired-prompt datasets collected independently of ViSU (e.g., MMA and SneakyPrompt). As a result, HyPE continues to perform consistently well across these settings, further validating that the model does not rely on superficial dataset-specific cues and instead generalizes robustly to unseen distributions.
>
> ---
> ---
>
>
> **Q2.**
> > Minor notation issue in Eq 1: if x is n+1 dimensional then the Lorentzian inner product should only sum up to n (the elements of x are indexed 0 to n, comprising n+1 elements).
>
> We clarify that in the Lorentzian inner product $\langle x, y \rangle_{\mathcal{L}} = -x_0 y_0 + \sum_{i=1}^{n+1} x_i y_i,$ the sum runs over the space-like components. Writing $n+1$ explicitly ensures all indices are accounted for, while $x_0 y_0$ is treated separately as the time-like component.
>
> ---
> ---
>
>
> **Q3.**
> > Suggest blurring the offensive image in Fig 1- readers will trust your claim that it is offensive.
>
> > Presentation: Table 1 misrepresents HyPE as the winning method in the Precision column for adv-MMA (should be GuardT2I). There are some ties that should also be marked- eg SneakyPrompt recall and COCO accuracy. Ideally you would also indicate statistically significant wins.
>
> We thank the reviewer for pointing this out. In the updated version of the paper, we provide a corrected version of the notation and added a more blurred version of Figure 1. We also provide a revised version of Table 1, correctly highlighting the winning methods.
>
> ---
> ---
>
>
> **Q4.**
> > For HyPS: How do you make the decision about what words to replace (is it a hard threshold or more adaptive method)?
>
> HyPS assigns positive anomaly scores to harmful words, while benign words receive scores close to zero or negative. In our method, we replace the top-1 or top-2 words with the highest scores, and we further analyze the effect of increasing the number of substituted tokens in **Figure 4b**. **Appendix A.4** provides illustrative word-cloud examples (top-1 and top-2), showing that HyPS consistently identifies genuinely harmful terms rather than relying on spurious correlations.
>
> ---
> ---
>
>
> **Q5.**
> > Is it difficult to implement LIG against the HySAC encoder?
>
> Implementing LIG on top of the HySAC encoder is not problematic. The integration path can be defined in the tangent space at the origin, and HySAC already provides stable and differentiable mapping functions that enable us to project points back and forth between the manifold and the tangent space.
> This makes the computation of integrated gradients tractable and does not introduce numerical instabilities in practice.
>
> ---
> ---
>
> **Q6.**
> > For HyPS experiments, do you also validate that the rewritten prompts fall inside R?
>
> The paper in Figure 4b illustrates this exact point, highlighting the trade-off between the number of sanitized words and the detection rate. The table shows that when the top 5 harmful words are substituted for safe ones, the majority of prompts fall inside the learnt R (between 65-85%), depending on the sanitization technique used.

---

> > ### Comment · Reviewer_WR4w · 2025-11-27
> > **Eq1 notation**
> >
> > Sorry to quibble on Q2, it is a minor point:
> >
> > If we agree first that $x \in \Re^2$ is 2-dimensional, then $x \in \Re^{n+1}$ is $(n+1)$-dimensional. As such the sum in Eq. 1 should count only $n+1$ terms but it clearly counts $n+2$, indexing the first term $x_0$ followed by a summation over $n+1$ terms indexed $(1, \ldots, n+1)$.  So there is a disagreement in notation between the definition of $x$ and the summation.

---

> > ### Comment · Reviewer_WR4w · 2025-11-27
> >
> > Thank you for clarifying and addressing my concerns. I have bumped by overall score to accept.

---

> ### Author Response · Authors · 2025-11-27
>
> We thank the reviewer for their thoughtful feedback and the helpful points raised during the rebuttal process. Following the suggestion, we have revised Equation 1 in the paper accordingly. We are glad our responses addressed their concerns, and we sincerely appreciate the positive evaluation of our work.
>
> We remain available for any further clarification.

---

### Official Review · Reviewer_kTKY · 2025-11-01

**Soundness:** 3
**Presentation:** 4
**Contribution:** 2
**Rating:** 4
**Confidence:** 2

**Summary:**

This paper proposes HyPE (Hyperbolic Prompt Espial) and HyPS (Hyperbolic Prompt Sanitization) to detect and mitigate harmful prompts for VLMs. HyPE models benign prompt embeddings with a hyperbolic one-class SVDD objective, leveraging hyperbolic geometry to better capture hierarchical semantic structures. HyPS then uses token-level attribution to locate malicious prompt components and sanitizes them while preserving intended meaning. Experiments on harmful prompt benchmarks show improved harmful-prompt detection and higher safety-preserving sanitization quality compared to Euclidean baselines and existing safety filters. Results cover multiple VLMs and attack settings.

**Strengths:**

1. Novel use of hyperbolic geometry for prompt safety detection; motivation around hierarchical semantics is clearly presented.
2. Covers both detection and sanitization, producing a more practical defense pipeline rather than only binary classification.
3. Experiments across multiple VLMs and datasets show consistent gains over Euclidean one-class and prompt-filtering baselines.

**Weaknesses:**

1. Evaluation focuses mainly on text-based harmful prompts; applying the method to multimodal triggers or image triggers would strengthen generality claims.
2. The robustness against paraphrased, or style-trigger [1] is unclear. Since harmful intent can be distributed across multiple tokens or advanced style rephrase, it is uncertain whether the method can consistently detect or sanitize such stealthy, paraphrased attacks. Evaluating these attack strategies would strengthen the claim of general robustness.

Ref:
[1] Mind the Style of Text! Adversarial and Backdoor Attacks Based on Text Style Transfer

**Questions:**

1. Can HyPE generalize to vision-triggered or multimodal jailbreak scenarios, beyond text-only attacks?
2. What is the runtime/memory overhead of hyperbolic SVDD compared to Euclidean baselines?
3. How robust is the method to paraphrased or iterative jailbreak prompts where harmful intent is spread across multiple tokens?

---

> ### Author Response · Authors · 2025-11-20
> **Detection on images**
>
> **Evaluation images**
> > Evaluation focuses mainly on text-based harmful prompts; applying the method to multimodal triggers or image triggers would strengthen generality claims.
>
> We thank the reviewer for the valuable comments and for recognizing that our work presents a novel and theoretically sound application of hyperbolic geometry, effectively proposing a full detection and sanitization pipeline for harmful prompts with consistent gains over Euclidean-based approaches.
>
> The primary goal of this paper is to implement a harmful-text detection and sanitization pipeline, aligned with state-of-the-art methods such as LatentGuard [a], DiffGuard [b], GuardT2I [c], and NSFW-Classifier [d].
>
> - [a] Liu et al. Latent guard: a safety framework for text-to-image generation, ECCV 2024.
> - [b] Gao et al. Diffguard: Semantic mismatch-guided out-of-distribution detection using pre-trained diffusion models, ICCV 2023.
> - [c] Yang et al. Guardt2i: Defending text-to-image models from adversarial prompts, NeurIPS 2024.
> - [d] https://huggingface.co/michellejieli/NSFW_text_classifier (**319,285** downloads last month on HuggingFace)
>
> We wish to emphasize that the central claim of this work is prompt-based detection that already has substantial and growing application potential. We will revise the manuscript to make this objective even clearer and ask the reviewer to reassess their evaluation in light of this stated focus.
>
> Nevertheless, since our embedding model is multimodal, we conducted a *preliminary experiment* to demonstrate the potential of extending our approach to image-based harmful content. The model was trained on 15k benign images generated from the ViSU training set, and the corresponding test set included both benign and malicious images following the same pipeline as in Poppi et al., 2025. We trained HSVDD on the benign images and compared its performance against FalconsAI [x], a widely adopted open-source NSFW image classifier available on HuggingFace.
>
> [x] https://huggingface.co/Falconsai/nsfw_image_detection
>
> | Model       | Precision | Recall  | F1 Score |
> |-------------|-----------|---------|----------|
> | **HyPE-images** | 0.5712    | 0.8773  | 0.6919   |
> | **FalconsAI**       | 0.9281    | 0.1187  | 0.2105   |
>
> These preliminary results show that HyPE-images can already detect malicious images effectively, particularly in terms of recall, despite being trained on only 10% of the available data. This indicates significant potential for future improvements and highlights the promise of extending our hyperbolic anomaly detection framework to multimodal harmful content. We will include these findings in the conclusion and outline future work to expand HyPE to image and multimodal scenarios.

---

> ### Author Response · Authors · 2025-11-20
> **Integration of style-attack**
>
> **Experiments with novel adaptive style-attack**
>
> We thank the reviewer for highlighting the importance of evaluating paraphrased or style-based attacks. We evaluated HyPE and four other models using StyleAttack [1] and conducted experiments on three datasets: ViSU, NSFW56k, and I2P*, across two attack settings with varying prompt paraphrasing strength (p).
> StyleAttack was executed independently for each model, with inference performed end-to-end. In this sense, the attack is adaptive and specifically tailored to each target model, making it a challenging test of robustness.
> For the ViSU dataset, we report precision, recall, and F1, consistent with our previous evaluations. For I2P* and NSFW56k, which are one-class datasets, we report the Attack Success Rate (ASR), defined as the number of times the attack successfully paraphrases a prompt to misclassify it as benign.
>
> The results demonstrate that HyPE consistently exhibits the highest robustness under these challenging conditions.
> In the p=0.4 setting, HyPE achieves the best performance on ViSU, with the second-best model being NSFW-classifier, while the remaining three models fail to detect harmful prompts.
> On I2P* and NSFW56k, HyPE achieves the lowest ASR, with gaps of 0.21 and 0.55 compared to the second-best model, confirming superior robustness against paraphrasing attacks.
> A similar pattern is observed at p=0.6, where HyPE continues to outperform NSFW-classifier on ViSU, while other models fail, and maintains the lowest ASR on I2P* and NSFW56k. These results indicate that even under previously unseen, adaptive, and style-based paraphrasing attacks, HyPE successfully detects harmful prompts. Detailed results are provided in **Appendix A.9**.
>
> - **p=0.4**
>
> |                     | **ViSU**          |              |              | **I2P\***      | **NSFW56k**    |
> |---------------------|-----------------|--------------|--------------|----------------|----------------|
> |                     | Pr ↑            | Rec ↑        | F1 ↑         | ASR ↓          | ASR ↓          |
> | NSFW-Classifier      | 0.65            | 0.65         | 0.65         | 0.72           | 0.82           |
> | DiffGuard            | 0               | 0            | 0            | 0.92           | 0.92           |
> | Detoxify (Orig)      | 0               | 0            | 0            | 1.0            | 0.91           |
> | Latent Guard         | 0               | 0            | 0            | 0.95           | 0.94           |
> | **HyPE (Ours)**      | **0.97**        | **0.67**     | **0.80**     | **0.51**       | **0.27**       |
>
> - **p=0.6**
>
> |                     | **ViSU**          |              |              | **I2P\***      | **NSFW56k**    |
> |---------------------|-----------------|--------------|--------------|----------------|----------------|
> |                     | Pr ↑            | Rec ↑        | F1 ↑         | ASR ↓          | ASR ↓          |
> | NSFW-Classifier      | 0.62            | 0.57         | 0.60         | 0.81           | 0.85           |
> | DiffGuard            | 0               | 0            | 0            | 0.93           | 0.92           |
> | Detoxify (Orig)      | 0               | 0            | 0            | 1.0            | 0.95           |
> | Latent Guard         | 0               | 0            | 0            | 0.97           | 0.97           |
> | **HyPE (Ours)**      | **0.97**        | **0.58**     | **0.73**     | **0.65**       | **0.32**       |

---

> ### Author Response · Authors · 2025-11-20
> **Questions**
>
> **Q1.**
> > Can HyPE generalize to vision-triggered or multimodal jailbreak scenarios, beyond text-only attacks?
>
> We thank the reviewer for this insightful question. While handling vision-triggered or multimodal jailbreaks is not the primary focus of this work, it represents an interesting and promising direction for future research. Our current study focuses on detecting harmful text prompts, which already covers a wide range of practical real-world applications, as discussed in the related work. Extending HyPE to multimodal scenarios could build on the same hyperbolic anomaly detection framework, but this remains beyond the scope of the present paper. We will revise the manuscript to include this as a future research direction.
>
> **Q2.**
> > What is the runtime/memory overhead of hyperbolic SVDD compared to Euclidean baselines?
>
> We thank the reviewer for this question. The runtime and memory overhead of HSVDD are essentially the same as those of Euclidean SVDD, as both methods share the same underlying architecture. The primary difference lies in the geometric formulation of the space, which does not introduce additional parameters or significant computational cost.
>
> **Q3.**
> > How robust is the method to paraphrased or iterative jailbreak prompts where harmful intent is spread across multiple tokens?
>
> We thank the reviewer for this insightful question and take the opportunity to clarify that robustness to paraphrased and iteratively obfuscated harmful prompts is already an integral part of our experimental evaluation. In fact, we further strengthened this component during the rebuttal phase by adding an additional attack.
> We evaluated HyPE on adversarially generated prompts specifically designed to mask or redistribute harmful intent through paraphrasing, token-level manipulation, or multi-step jailbreak procedures. In particular, we tested against two state-of-the-art jailbreak and obfuscation attacks, namely:
>
> 1. MMA-Diffusion: Multimodal attack on Diffusion Models, CVPR 2024
> 2. Sneakyprompt: Jailbreaking text-to-image generative models, IEEE symposium on security and privacy 2024.
>
> In addition, for the rebuttal, we incorporated a new adaptive paraphrasing attack, the StyleAttack of Qi et al., which generates stylistically transformed prompts that remain semantically aligned with the harmful intent while attempting to evade detection. As discussed in **Appendix A.9 (Tables 8-9)**, HyPE consistently outperforms all competing methods across these attack settings.
> We thank the reviewer for the valuable question and take the opportunity to clarify that this set of experiments is already included in our evaluation, and has been further strengthened during the rebuttal phase with the addition of an extra attack.
>
> We hope that the additional clarifications and complementary experiments will contribute to a more positive reassessment of our contribution.

---

### Official Review · Reviewer_9tyq · 2025-11-02

**Soundness:** 3
**Presentation:** 3
**Contribution:** 3
**Rating:** 6
**Confidence:** 2

**Summary:**

The paper proposes HyPE (Hyperbolic Prompt Espial) and HyPS (Hyperbolic Prompt Sanitization), two complementary modules for detecting and sanitizing harmful prompts in Vision–Language Models (VLMs). HyPE employs a hyperbolic SVDD anomaly detector trained solely on benign prompts to identify harmful inputs as outliers in Lorentz space, while HyPS applies explainable attribution (Layer Integrated Gradients) to localize harmful words and neutralize them through replacement or removal, preserving semantics. Experiments across six datasets, multiple adversarial attacks, and two downstream tasks (text-to-image generation and image retrieval) demonstrate superior detection accuracy, interpretability, and robustness over prior methods .

**Strengths:**

- The paper extends classical SVDD to hyperbolic space (Eq. 2) using the Lorentz distance and curvature parameter K. This geometric formulation enhances interpretability and robustness, which are often lacking in prior NSFW classifiers.
- The one-class setup removes the need for harmful data training, improving safety and scalability, an important property since unsafe content may not always be known or accessible to defenders.
- The method is evaluated on six datasets spanning paired and single-class prompts as well as adversarial attack settings. Results show consistent improvements over multiple baselines, demonstrating strong practical effectiveness.

**Weaknesses:**

- The proposed approach appears to be a direct adaptation of SVDD into Lorentz space, with limited new theoretical insights. For instance, the Eq. (2) largely represents a straightforward geometric reformulation rather than a fundamentally new concept.
- HyPE relies on embeddings from HySAC, which may introduce bias and limit the generality of reported performance.
- All evaluated datasets focus on English NSFW and violent content, with no assessment of multilingual or socio-cultural harmful expressions, potentially constraining generalization.

**Questions:**

Could the authors provide additional experiments or discussion on multilingual settings and other categories of safety violations to support the broader applicability of their framework?

---

> ### Author Response · Authors · 2025-11-20
> **Advantages of HSVDD**
>
> **Comparison betweenn SVDD and HSVDD**
> > The proposed approach appears to be a direct adaptation of SVDD into Lorentz space, with limited new theoretical insights. For instance, the Eq. (2) largely represents a straightforward geometric reformulation rather than a fundamentally new concept.
>
> We sincerely thank the reviewer for their feedback and constructive comments. We are glad that the reviewer acknowledges that our approach effectively addresses the problem of harmful prompt detection, showing consistent improvements over the proposed baselines. We also appreciate the recognition that one strong contribution of our work is the ability to rely solely on safe data, addressing the challenge of scarce unsafe data. Additionally, the reviewer highlights that our method provides interpretable predictions, which are often lacking in other approaches.
>
> The main contribution of our work is to frame harmful prompt detection as an anomaly-detection task in hyperbolic space. This formulation allows HyPE to be trained exclusively on benign data, avoiding exposure to harmful content. The essence of our contribution regarding the adaptation of SVDD to Lorentz space originates from a preliminary analysis of the clustering of safe and harmful prompts in the embedding space. Visualizations using UMAP, now included in Appendix A.3-Figure 9, show that prompts follow a hierarchical structure consistent with the intrinsic properties of hyperbolic space and guided by the entailment-based loss.
> We observed that embeddings naturally organized themselves along this hierarchical structure, which aligns well with the anomaly-detection task and the modeling capabilities of SVDD. However, traditional Euclidean SVDD approaches were insufficient to capture this structure (as depicted in Figure 9 and quantitatively evaluated in Table 5-6).
>
> To address this, we reformulated SVDD in hyperbolic space, enabling the model to learn an arbitrarily centered subregion of the hyperboloid enclosed within a hyperbolic sector, thereby developing our novel HSVDD algorithm.
> Furthermore, unlike Euclidean SVDD, which must learn both the center and the radius of the hypersphere, HSVDD does not require learning the center because the hyperbolic embedding space naturally organizes benign prompts near the origin. This design reduces the number of trainable parameters to a single lightweight parameter R, while leveraging the inherent structure of hyperbolic space.
> The following table (listed **Table 7** in the revised manuscript) reports the detection performance of SVDD in Euclidean space compared to our HSVDD in hyperbolic space. As observed, the performance gap between the two approaches is substantial, exceeding 90% and 32% in precision and F1, respectively.
>
> | Method      | Pr    | Rec   | F1    |
> |-------------|-------|-------|-------|
> | CLIP-SVDD   | 0.08  | 0.96  | 0.66  |
> | Hype        | **0.98** | **0.98** | **0.98** |
>
>
> The novelty of our approach goes beyond the hyperbolic encoder itself. We selected HySAC, a hyperbolic encoder explicitly trained on benign and harmful data, as our foundation, since publicly available alternatives are limited and retraining is not feasible. We acknowledge that relying on HySAC embeddings may introduce bias and limit the generalizability of reported performance. This limitation is now clearly discussed in the revised manuscript and highlighted as a direction for future research.

---

> ### Author Response · Authors · 2025-11-20
> **Bias and dataset composition**
>
> **Model bias**
> > HyPE relies on embeddings from HySAC, which may introduce bias and limit the generality of reported performance.
>
> The proposed approach adopts HySAC as the hyperbolic encoder, which is currently the only hyperbolic encoder explicitly trained on harmful data and represents the state of the art for multiple downstream tasks [1], including generation and retrieval. We are confident that our work further highlights the value of hyperbolic geometry in this domain and can encourage its broader application in related tasks. We have revised the manuscript to clarify our integration with HySAC and explicitly acknowledge, in the conclusion, the need to further develop hyperbolic encoders for safety-critical applications. We believe our work reinforces the potential of this framework and points to promising directions for future research.
>
>
>
> **Dataset heterogeneity**
> > All evaluated datasets focus on English NSFW and violent content, with no assessment of multilingual or socio-cultural harmful expressions, potentially constraining generalization.
>
> All evaluated datasets focus on English NSFW and violent content, with no assessment of multilingual or socio-cultural harmful expressions, potentially constraining generalization.
> We wish to clarify to the reviewer that our evaluation, in addition to paired datasets where each benign prompt has a corresponding malicious prompt, already includes datasets containing only benign prompts. The proposed datasets differ significantly from one another, covering a wide range of harmful content categories, from pornography to violence. In particular, the ViSU dataset, as shown in [1], was designed to encompass diverse harmful possibilities.
> Including fully benign datasets, such as COCO, further demonstrates that HyPE maintains strong performance when facing non-harmful prompts. **Appendix A.2** presents the datasets in more detail and provides examples of prompts within them.
>
>
> [1] Poppi et al. HySAC: Hyperbolic Safety-Aware Vision-Language Models, CVPR 2025.

---

> ### Author Response · Authors · 2025-11-20
> **Multilingual setting**
>
> **Comparison in the multilingual setting**
>
> To assess generalization to multilingual inputs, we extended our evaluation with three translated versions of the ViSU dataset in Spanish, Italian, and French. Results reported in **Appendix A.8** (and the table below) show that HyPE consistently maintains state-of-the-art performance across these languages. Overall, these experiments demonstrate that HyPE is robust not only across different harmful categories but also across benign content and multiple languages, supporting the generalization of our approach.
>
> |                     |       |    **ViSU-sp** |       |       |      **ViSU-fr**       |       |       |     **ViSU-it**        |       |
> |---------------------|-----------------|-------------|-------|-----------------|-------------|-------|-----------------|-------------|-------|
> |                     | Pr              | Rec | F1    | Pr              | Rec         | F1    | Pr              | Rec         | F1    |
> | NSFW-Classifier      | 0.59 | 0.58        | 0.58  | 0.72           | 0.43        | 0.54  | 0.70           | 0.38        | 0.49  |
> | DiffGuard            | 0.92 | 0.15        | 0.25  | 0.81           | 0.20        | 0.32  | 0.92           | 0.12        | 0.21  |
> | Detoxify (Orig)      | **0.96** | 0.07        | 0.14  | **0.99**       | 0.08        | 0.14  | **0.97**       | 0.08        | 0.14  |
> | Latent Guard         | 0.65 | 0.38        | 0.48  | 0.64           | 0.25        | 0.36  | 0.60           | 0.50        | 0.54  |
> | **HyPE (Ours)**      | 0.73 | **0.90**    | **0.81** | 0.75        | **0.87**    | **0.81** | 0.78        | **0.84**    | **0.81** |
>
>
> We remain available for any additional requests or clarifications. Meanwhile, we hope that these additional experiments and clarifications will further convince the reviewer of the value of our work and may encourage them to increase their assessment on our contribution.

---

> > ### Comment · Reviewer_9tyq · 2025-11-26
> >
> > Thanks for the detailed response. It is much appreciated.
> >
> > > The essence of our contribution regarding the adaptation of SVDD to Lorentz space originates from a preliminary analysis of the clustering of safe and harmful prompts in the embedding space.
> >
> > This explanation helps clarify the contribution of the work.
> >
> > > Comparison in the multilingual setting
> >
> > This addition makes the experimental evaluation more comprehensive.
> >
> > Overall, all my questions have been addressed, and I remain positive toward this work.

---

> > > ### Author Response · Authors · 2025-11-27
> > >
> > > Thank you again for your thoughtful follow-up and for the time you devoted to reviewing our work. We appreciate your positive remarks, and we are glad that the additional analyses helped clarify our contributions and fully addressed your questions.
> > > Since we are still within the revision window, please let us know if any further refinements could strengthen the submission.
> > > We would be more than happy to incorporate any final suggestions you might have.
> > >
> > > We also note that a rating of 6 is described as "marginally above the acceptance threshold, **but would not mind if the paper were rejected**".
> > > If at this stage you feel comfortable with the value of the work as presented, and believe it merits stronger support, we would be grateful for any *adjustment* you deem suitable. Otherwise, any indication of what could still be improved would be equally appreciated.
> > >
> > > Thank you again for your careful assessment and constructive engagement throughout the process.

---

### Official Review · Reviewer_qMAf · 2025-11-03

**Soundness:** 3
**Presentation:** 3
**Contribution:** 2
**Rating:** 4
**Confidence:** 3

**Summary:**

This paper proposes a hyperbolic geometry–based framework to detect and neutralize harmful prompts in VLMs. It introduces two modules: HyPE, a hyperbolic anomaly detector that models benign prompts in hyperbolic space and flags harmful ones as outliers, and HyPS, a post-hoc sanitization system that identifies harmful words and replaces or modifies them using lookups or LLM-based substitution. Experiments across six datasets and two adversarial attacks show that HyPE outperforms prior detectors in accuracy and robustness, while HyPS effectively sanitizes malicious prompts without distorting their meaning.

**Strengths:**

1. HyPE demonstrates the SOTA detection performance on six datasets and two adversarial scenarios
2. HyPS performs well on two downstream tasks

**Weaknesses:**

Main concern
1. The contribution is relatively incremental. This paper mainly applies hyperbolic models to harmful prompt detection in VLMs. Apart from the empirical gains shown in the experiments, more analysis of why we need to use the hyperbolic models in this task is needed (e.g., some theoretical insights or some empirical comparison between the SVDD and hyperbolic SVDD, etc).

Other concerns
1. The authors do not consider the adaptive attack scenario, in which the attackers have white-box access to both the detectors and the text encoder.
2. The authors only evaluate the effectiveness of one text encoder, HySAC. More evaluations on other encoders would show the robustness of this approach.

**Questions:**

1. If the classifier were not based on a hyperbolic model, would HePS still function effectively? Does this module have to depend on hyperbolic modeling?

---

> ### Author Response · Authors · 2025-11-20
> **Justification of hyperbolic modeling**
>
> **Contribution and justification over HSVDD**
>
> We thank the reviewer for the constructive feedback. To address this concern, we have revised the paper to include a detailed comparison and visualization demonstrating why HSVDD outperforms standard SVDD. UMAP projections reveal that the embedding space of benign and malicious prompts exhibits a hierarchical structure that is more naturally captured in hyperbolic space. Detection in this space becomes more effective, as also supported by the quantitative results in **Tables 5**  and **7** (**Appendix A.3**) and illustrative results in **Figure 9** with a 3D UMAP projection of harmful and benign prompts.
> We also report in the following table (listed **Table 7** in the revised paper) a quantitative comparison between the proposed HyPE approach and a trained Euclidean-based SVDD anomaly detector that uses CLIP as its embedding backbone. As observed, the performance gap between the two approaches is substantial, exceeding 90% and 32% in precision and F1, respectively.
>
> | Method      | Pr    | Rec   | F1    |
> |-------------|-------|-------|-------|
> | CLIP-SVDD   | 0.08  | 0.96  | 0.66  |
> | Hype        | **0.98** | **0.98** | **0.98** |
>
> Furthermore, HSVDD provides a generalization advantage as its optimization is framed as an anomaly detection problem with respect to the benign class, making it more sensitive to malicious prompts, which typically exhibit anomalous behavior in the embedding space (as clearly visible in the UMAP projections in **Figure 9**  in the revised manuscript). Contrary, prior methods (e.g., LatentGuard and GuardT2I) model detection as learning an arbitrary-shaped classification function in the latent space. Such approaches often show dataset-dependent performance, as reflected in **Table 1**. The hyperbolic formulation of HSVDD captures the latent space structure more faithfully, reflecting the inherent hierarchy and geometry of the embeddings, which translates into more robust and consistent detection across diverse datasets.
>
> We hope that this clarification, along with the new comparisons and visualizations, now addresses the reviewer’s main concern.

---

> ### Author Response · Authors · 2025-11-20
> **Implementation of adaptive attacks**
>
> **Adaptive attack scenarios**
>
> We thank the reviewer for highlighting the importance of evaluating adaptive attacks. To address this, we have added two new adaptive attack scenarios in the revised manuscript.
> The first is an adaptive version of the StyleAttack from Qi et al. [1] (presented in **Appendix A.9**), which paraphrases prompts to evade detection. The attack was applied independently to each model, including ours. The second Yang et al. [2] is our custom adaptive attack specifically designed to challenge HyPE, formulated in **Section 4.1**. The results show that HyPE remains highly robust under these adaptive attacks.
> Specifically, in the StyleAttack evaluation on the ViSU, NSFW56k, and I2P* datasets, HyPE consistently outperforms all baselines. For ViSU, precision, recall, and F1 are significantly higher than competing models, and on I2P* and NSFW56k, the Attack Success Rate is substantially lower (i.e., better detection rate from HyPE), with gaps of 0.21 and 0.55 compared to the second-best model. Even at higher paraphrasing strength (p=0.6), HyPE maintains strong detection performance while other models fail. These results demonstrate that HyPE reliably detects harmful prompts, even under challenging adaptive paraphrasing attacks.
> Notably, the custom white-box adaptive attack provides further insight into the limitations of evading HyPE. To construct this attack, we introduced a parameter $\lambda$ in the adaptive penalty term of Eq. 5, which controls the importance of the attacker’s objective to evade detection. As $\lambda$ increases from 0 to 1, the attack progressively prioritizes evading HyPE by generating adversarial prompts that lie within the anomaly region of radius R.  The results obtained in Table 3 of the revised manuscript are really interesting. When $\lambda$ is small, HyPE continues to detect adversarial prompts, as a natural consequence of the fact that the attacker is not giving enough importance during the optimization to break the defense. Oppositely, as $\lambda$ approaches 1, detection fails. However, surprisingly, we can notice that the prompts themselves undergo drastic structural changes and no longer preserve malicious intent (**Appendix A.10, Table 10**, and **Figure 16**). This indicates that in order to evade HyPE, an attacker must effectively remove the harmful content from the prompt. In practice, this means that adaptive attacks cannot succeed without compromising the original malicious intent. **Figure 16** illustrates how the generated images from Stable Diffusion change across different levels of $\lambda$, and **Table 10** reports an example of prompt evolution when increasing $\lambda$. At higher values of $\lambda$, the prompts are severely less harmful, and the resulting images and prompts no longer reflect the original malicious intent, visually confirming that evading HyPE requires removing or drastically diminishing the harmful intent.
>
> Overall, these evaluations show that HyPE not only withstands standard attacks but also maintains strong robustness under adaptive scenarios, providing a meaningful improvement over prior methods. We believe this finding is important and novel, as it highlights an interesting tradeoff between adaptive attacks and the preservation of malicious intent in prompts.
>
> We hope it further convinces the reviewer of the novelty and significance of our proposal.
>
> [1] Qi et al. Mind the Style of Text! Adversarial and Backdoor Attacks Based on Text Style Transfer. EMNLP 2021
>
> [2]Yang et al.  MMA-Diffusion: Multimodal attack on Diffusion Models, CVPR 2024.

---

> ### Author Response · Authors · 2025-11-20
> **Text encoder and questions**
>
> **Single text encoder**
> > The authors only evaluate the effectiveness of one text encoder, HySAC. More evaluations on other encoders would show the robustness of this approach.
>
> We wish to clarify to the reviewer that the proposed approach is grounded in hyperbolic geometry, and the application of this framework to harmful prompt detection is a recent development in the literature [1] [2]. As a result, there is a limited number of encoder models based on hyperbolic embeddings. Among the available options, HySAC is currently the only open-source model explicitly trained on benign and harmful data, and it has been shown to achieve state-of-the-art performance in embedding space modeling. As demonstrated in its original work, HySAC provides an accurate foundational model that enables the development of multiple downstream task pipelines.
> We believe our work reinforces the potential of this framework and highlights the need to further develop hyperbolic encoders for safety-critical applications. We have revised the manuscript to explicitly acknowledge this limitation and discuss possible directions for future research.
>
> [1] Poppi et al. HySAC: Hyperbolic Safety-Aware Vision-Language Models, CVPR 2025.
>
> [2] Pal et al. Compositional entailment learning for hyperbolic vision-language model, ICLR 2025.
>
>
> **Questions**
> > If the classifier were not based on a hyperbolic model, would HePS still function effectively? Does this module have to depend on hyperbolic modeling?
>
> HyPS is a module that leverages a post hoc explanation method, namely LIG (LayerIntegratedGradients), and for this reason, it can be easily integrated and work with any differential distance-based anomaly Detector, since it relies on the model’s predictions (even ones operating in the Euclidean space). We introduced it to add interpretability to our detector and to analyze whether its decisions were based on spurious correlations or not. We then adopted its attributions to guide the sanitization.

---

> ### Comment · Reviewer_qMAf · 2025-11-27
>
> The additional experiments and explanation address my main concerns about the novelty and robustness, and I would like to raise my score.

---

> > ### Author Response · Authors · 2025-11-27
> >
> > Dear reviewer, we are glad that the additional analyses helped clarify our contributions and fully addressed your questions, and we appreciate your positive re-evaluation.
> >
> > Since we are still within the revision window, please let us know if any further refinements could strengthen the submission. We would be happy to incorporate any final suggestions that might help you feel more confident about the value of our work.

---

### Author Response · Authors · 2025-11-20
**Summary of the rebuttal**

We thank the reviewers for acknowledging our contributions and for providing constructive feedback, which has substantially improved the quality and clarity of our work. In particular, after the rebuttal, the paper now:

- **Provides a more in-depth justification for hyperbolic geometry**, including new UMAP visualizations and an extended empirical comparison showing that HSVDD significantly outperforms Euclidean SVDD in precision, recall, and F1.

- **Adds two adaptive attack evaluations**, including a custom white-box adaptive attack and an adaptive version of StyleAttack, demonstrating that HyPE remains robust even under adversarial paraphrasing and targeted evasion.

- **Extends the evaluation to multilingual scenarios** (Spanish, Italian, French) using translated ViSU datasets, showing strong zero-shot cross-language generalization.

- **Fixes presentation issues**, including the proper blurring of harmful images.

We remain available to the reviewers for any further clarification or additional analysis if needed.

---

> ### Author Response · Authors · 2025-11-26
> **Summary of updates to the paper**
>
> We are very grateful to the reviewers for the insightful feedback we received, and we would like to take this opportunity to thank them once again.
>
> We would also like to point out that the following parts of the paper have been modified:
>
> 1. We blurred and modified the NSFW image in the teaser figure.
> 2. In the main paper, we added the paragraph "Adaptive Attacks" in Section 4.1 (Experimental Setup) and the corresponding paragraph "Adaptive Attack" in Section 4.2 (Experimental Results).
> 3. We also added Table 3 and Figure 8, which illustrate the performance of HyPE under the worst-case adaptive attack and show images generated from the obtained adversarial prompts for $\lambda = 0.1$ and $\lambda = 1$, respectively.
> 4. We updated Appendix A3 by adding a justification for opting out of hyperbolic geometry.
> 5. We added three completely new sections in the appendix (A8, A9, and A10), containing experiments on multilingual transferability, adaptive StyleAttack, and the adaptive attack, respectively.
>
> We kindly invite reviewers to consider these revisions, which represent our direct response to their feedback and our commitment to improving the paper in line with their suggestions.

---

### Author Response · Authors · 2025-11-30
**Letter to the AC, summary of the rebuttal (pt 1)**

Dear Area Chair,

We would like to summarize the overall quality of our submission, the contribution recognized by the reviewers, and the outcome of the rebuttal period, which has significantly strengthened the paper and the confidence and rating from the reviewers in it.


First, the reviewers acknowledged the quality, novelty, and practical relevance of our approach. The core ideas of leveraging hyperbolic geometry for harmful prompt detection, relying only on benign data, and coupling detection with a sanitization module were consistently highlighted as meaningful contributions. Several reviewers emphasized that the hyperbolic formulation is well-motivated, interpretable, and practically valuable for mitigating harmful prompts.

> The paper extends classical SVDD to hyperbolic space (Eq. 2) using the Lorentz distance and curvature parameter K. This geometric formulation enhances interpretability and robustness, which are often lacking in prior NSFW classifiers.


> The additional experiments and explanation address my main concerns about the novelty and robustness, and I would like to raise my score.

> The one-class setup removes the need for harmful data training, improving safety and scalability, an important property since unsafe content may not always be known or accessible to defenders.

> Novel use of hyperbolic geometry for prompt safety detection; motivation around hierarchical semantics is clearly presented.

> Covers both detection and sanitization, producing a more practical defense pipeline rather than only binary classification.

> the hyperbolic approach is well-suited to single-class training, reducing a dependency on labeled malicious examples.

> The method includes a sanitization step, enabling end-to-end workflows where refusal is undesirable.

Second, the reviewers recognized that our method achieves state-of-the-art performance and has a clear impact. Beyond the original version of the paper, the rebuttal period enabled us to substantially expand the experimental evidence.

> Empirical experimental results are relatively thorough and demonstrate strong performance of the method compared to several baselines.

> This addition makes the experimental evaluation more comprehensive.

We introduced new comparisons between Euclidean SVDD and Hyperbolic SVDD, added multilingual evaluations, clarified the generalization properties arising from dataset heterogeneity, and integrated two adaptive attacks, including a custom white-box attack and StyleAttack at multiple perturbation levels. Our method, HyPE, consistently performed best under these challenging conditions.
Furthermore, although multimodal protection was not the intended scope of the paper, we also provided preliminary experiments showing that extending our framework to image-based harmful content yields even better performance than FalconsAI, a widely used NSFW classifier on HuggingFace with over 67 million downloads in the last month alone (https://huggingface.co/Falconsai/nsfw_image_detection). This demonstrates the generality of our method well beyond its original design.

---

> ### Author Response · Authors · 2025-11-30
> **Letter to the AC, summary of the rebuttal (pt 2)**
>
> Third, and most importantly, the rebuttal process clearly improved the assessments and confidence from reviewers.
>
> > Thank you for clarifying and addressing my concerns. I have bumped by overall score to accept.
>
> > The additional experiments and explanation address my main concerns about the novelty and robustness, and I would like to raise my score.
>
> > Overall, all my questions have been addressed, and I remain positive toward this work.
>
>
>
>
>
> **Reviewer qMAf** (score 4 --> 6, high confidence) was initially concerned about the novelty of HSVDD and the need for stronger adaptive attack evaluations. Our additional experiments and clarifications directly addressed these concerns. The reviewer explicitly increased their score while maintaining high confidence and wrote: "The additional experiments and explanation address my main concerns about the novelty and robustness, and I would like to raise my score". The Area Chair can indeed see Appendix A3, which now includes a justification for the need for hyperbolic geometry, as well as the corresponding paragraph "Adaptive Attack" in Section 4.2 (Experimental Results). These paragraphs and analysis directly respond to this reviewer, highlighting factual improvements in the paper.
>
> **Reviewer 9tyq** (score 6 --> unknown),  was already positive with a rating of 6. After reading our rebuttal and revised paper, they expressed full satisfaction, citing the improved comparison between SVDD and HSVDD, the clearer motivation for hyperbolic geometry, the explanation of dataset heterogeneity, and the added multilingual experiments. Their comment concluded: "Overall, all my questions have been addressed, and I remain positive toward this work". Although score editing was closed by then, the content of their follow-up strongly suggests that their evaluation improved.
>
> **Reviewer kTKY** (score 4, low confidence) did not engage during the rebuttal. Their review contained only two short, minor concerns. We addressed both: we clarified that our scope is text-only (as in many prior works), yet still provided new multimodal results that surpass FalconsAI, and we integrated the StyleAttack evaluation into the revised paper. Even though no follow-up was posted, these changes fully resolve the issues raised in this review. Specifically, regarding the first one, although multimodality was not our original aim, we still went beyond the scope of the paper and conducted preliminary experiments that extended our approach to vision-based harmful content. These experiments showed that our method outperforms FalconsAI, a widely adopted NSFW image classifier on HuggingFace with over 67 million downloads in the last month. We consider this a strong additional result, given that it was not originally part of the work. Regarding the second one, we included the additional style attack in the rebuttal and in the revised paper (presented in Appendix A.9), evaluating StyleAttack at different perturbation levels. In both settings, HyPE demonstrated the highest robustness among all the related works and setups.
>
> Lastly, **Reviewer WR4w** (score 6 --> 8, high confidence) requested more theoretical and practical justification for Hyperbolic SVDD over the Euclidean variant. Our additional responses, analysis, and experiments clarified these points, and the reviewer increased their score from 6 to 8, maintaining high confidence. We also corrected a minor typo highlighted during our discussion.
>
> In summary, only one low-confidence reviewer did not participate, though our responses directly addressed their feedback. Two high-confidence reviewers increased their scores after the rebuttal, and a third reviewer was fully satisfied and remained open to increase the rating.
> We believe these outcomes reflect the solidity, novelty, and impact of the work, as well as the substantial improvements incorporated during the rebuttal. The rebuttal has been conducted in a sound, ethical, and constructive manner, and all evidence from the discussions has led to real and concrete changes in the revised paper (see the comment "Summary of updates to the paper" for further details).
> The discussion with many reviewers was constructive, and we are pleased that the quality of our work has improved as a result of our interaction with them.
>
> We appreciate your consideration and remain available for any clarification.

---

### Meta-Review · Area_Chair_cNvD · 2026-01-05

**Summary:**

The paper introduces HyPE and HyPS, a framework for detecting and sanitizing harmful prompts in VLMs by leveraging hyperbolic geometry. The core insight is that benign and harmful prompts exhibit a hierarchical structure in embedding space that is better captured by hyperbolic geometry (Lorentz model) than Euclidean space. Reviewers initially recognized the novelty of the geometric approach but questioned its practical necessity compared to standard Euclidean baselines and its robustness against adaptive attacks. The authors provided a highly effective rebuttal, including a direct comparison showing massive gains over Euclidean SVDD, new adaptive attack evaluations (StyleAttack and custom white-box), and multilingual experiments. Following these additions, the active reviewers significantly raised their scores/confidence.

**Reviewer Concerns:**

1. Necessity of Hyperbolic Geometry: The reviewers questioned if the complexity of hyperbolic space was justified. The authors provided a direct comparison where HyPE (Hyperbolic SVDD) achieved an F1 of 0.98 versus 0.66 for Euclidean SVDD, alongside UMAP visualizations showing the hierarchical separation. This completely resolved the concern.

2. Adaptive & Stealthy Attacks (Reviewers qMAf, kTKY): Concerns about the lack of adaptive attacks were addressed by adding a custom white-box attack (showing attackers must remove harmful content to evade detection) and StyleAttack (paraphrasing), where HyPE outperformed baselines.

3. Generalization: The authors addressed concerns about limited scope by adding multilingual experiments (Spanish, French, Italian) and a preliminary image-detection experiment that outperformed a standard baseline (FalconsAI).

4. Notation and Presentation (Reviewer WR4w): Minor notation errors in the Lorentz product formula were fixed, and sensitive images were blurred.

**Reviewer Scores:**

Reviewer WR4w: 6 -> 8 (Explicitly raised score).

Reviewer qMAf: 4 -> 6 (Explicitly raised score).

Reviewer 9tyq: 6 (Maintained positive stance, confirmed all questions answered).

Reviewer kTKY: 4 -> 4 (maintain)

---

### Decision · Program_Chairs · 2026-01-26

Accept (Poster)